# Effect of Amylose and Crystallinity Pattern on the Gelatinization Behavior of Cross-Linked Starches

**DOI:** 10.3390/polym14142870

**Published:** 2022-07-15

**Authors:** Tingting Kou, Jun Song, Mouquan Liu, Guihong Fang

**Affiliations:** 1College of Food Engineering and Biotechnology, Hanshan Normal University, Chaozhou 521041, China; ktings@163.com (T.K.); liumouquan@163.com (M.L.); 2Center for Biomedical Optics and Photonics (CBOP) & College of Physics and Optoelectronic Engineering, Key Laboratory of Optoelectronic Devices and Systems, Shenzhen University, Shenzhen 518060, China; songjun@szu.edu.cn; 3Department of Nutrition and Food Hygiene, Hainan Medical University, Haikou 571199, China; 4Carbohydrate Laboratory, School of Food Science and Engineering, South China University of Technology, Guangzhou 510640, China

**Keywords:** cross-linked starch, pasting properties, amylose content, crystalline pattern

## Abstract

Starches from normal maize (NM), normal potato (NP), waxy maize (WM), and waxy potato (WP) were cross-linked with seven different concentrations (0.01, 0.05, 0.1, 0.5, 1, 5, 10%) of sodium trimetaphosphate and sodium tripolyphosphate. The use of low-amylose WM and WP as well as A-crystalline maize and B-crystalline potato starches can determine the influence of the amylose content and crystallinity pattern on the cross-linking of starches. The results showed that the viscosity of the cross-linked starch (CLs) first increased and then deceased, and finally no viscosity was detected; WM showed no viscosity at 5% and NP at 1%. In addition, the viscosity of NM first increased and then became undetectable at 0.5%. Strikingly, the WP developed viscosity even at a 10% reagent level (RL), and it developed the highest viscosity of all samples at 1%. The starch-iodine method was a facile and high-performance method for the characterization of the cross-linking degree (CL%), having been applied to normal starches, because the increase in the CL% resulted in a decrease of iodine-complexed amylose and blue intensity. In this study, the starch-iodine method was extended to waxy starches, which stained brown with iodine, and the brown intensity decreased with the increase of the CL%. Moreover, the CL% and RL showed a linear-log relationship.

## 1. Introduction

Starches from different genetic mutations of a plant are mainly classified into three different groups: waxy, normal, and amylose extender type [1,2,3,4]. Amylopectin takes up almost the entire content of waxy starch and is also the main component of normal starch [2,4,5]. Waxy and normal starches can develop high viscosities when gelatinized in aqueous solutions, and are widely used as a filling, thickening, or stabilizing agent in the food industry [6], as a surface sizing agent in the papermaking industry [7], and as a highly viscous nanogel for oil recovery [8]. However, it is difficult for high amylose starch to develop viscosity, it contains more amylose than other starches and has been found to be more prone to retrograde [5,9,10]. As a result, the high amylose starch is usually found in the application of non-thickening industries such as dietary fiber [9,11], emulsion stabilization [12,13], and drug delivery [14].

Cross-linking reagents can bind neighboring anhydro-glucose units in the amorphous regions of amylose and amylopectin in the normal starch, and amylopectin in the waxy starch [15]. Chemical, physical, and ionic cross-linking are the three most common ways developed for cross-linking [16], and chemical cross-linking is the most widely available and effective strategy by its stronger bonds [17]. Wei et al. [18] applied epichlorohydrin cross-linked porous corn starch to enhance its mechanical property and adsorption capacity. Lu et al. [19] prepared borax cross-linked starch nanoparticles to reinforce the starch films, which displayed an improved tensile strength, toughness opacity, and water vapor permeability. Peidayesh et al. [20] reported citric acid cross-linked corn starch and chitosan hydrogel, which showed good water resistance and gel content. With regards to the high efficiency, a 0.01% STMP/STPP reagent level could significantly increase the final viscosity [21]. Cross-linking helped in reducing the moisture absorption, and the cross-linked WM was used to obtain smooth, transparent, and defect-free films comparable to commercially available plastic sheets [22].

In the past, CLs have been prepared for various applications, these include thickeners, drug delivery material, and other composite materials [23,24,25]. In this paper, we compared the differences between cross-linked normal starch and waxy starch, and A-crystalline maize starch and B-crystalline potato starch, using STMP/STPP as the cross-linker, which is widely used in many industries. After cross-linking, amylose–amylopectin and amylopectin–amylopectin linkages were constructed in normal starches, and amylopectin–amylopectin linkages in waxy starches [15]. The amylose helix provided a tunnel for the iodine molecules to align, and the long branch chains of amylopectin could also complex with iodine and give additional iodine affinity [26]. After being cross-linked to amylopectin, the blue intensity of the amylose-containing starch–iodine decreased and a facile method to characterize the CL% was proposed, i.e., the starch–iodine (St-I) method [21]. Until recently, waxy starches were not included. In this study, St-I was used to determine the CL% of waxy starch and compared with the conventional viscosity method. In addition, the effects of the amylose content and crystallinity were also discussed.

## 2. Materials and Methods

### 2.1. Materials

All the starches used in this study were of food-grade, normal, and waxy starches of A and B crystalline patterns. NM and WM (A type) were provided by Chemical Technology Co., Ltd. (Dongguan, China), and NP and WP (B type) were provided by the Dutch Meelunie Company (Amsterdam, The Netherlands). Chemicals and solvents were at least of an analytical grade. The amylose content of the four starches were determined by the ISO6647-1:2007 [27], i.e., NM 31%, WM 0%, NP 36%, and WP 3.4%.

### 2.2. Preparation of Cross-Linked Starch (CLs)

CLs of NM and NP were the samples prepared in a previously published paper [21], and CLs of WM and WP were prepared according to the method reported in that paper. In brief, starch (20 g, dry base), water (80 mL), and sodium sulfate (2.0 g) were mixed in an Erlenmeyer flask. The mixture was adjusted to pH 11.5 with 1.0 M sodium hydroxide and then the cross-linking reagent (STMP: STPP = 99:1) of different levels (0.01, 0.05, 0.1, 0.5, 1, 5, and 10% of the dry starch) were added. The slurry was stirred continuously, warmed up to 45 °C, and held at this temperature for 3 h. Then the slurry was adjusted to pH 6.5 by adding 1.0 M hydrochloric acid, the starch was collected by centrifugation (3000× *g*, 10 min), washing with water, and dried at 45 °C overnight.

### 2.3. X-ray Diffraction (XRD)

XRD patterns were obtained according to a previously reported method and relative crystallinity was determined by the ratio of the peak areas to the total diffractogram area [4,28].

### 2.4. Light Microscopy

Light microscopy under normal light and polarized light of native and their CLs (10% RL) was analyzed according to a method described previously [4].

### 2.5. Differential Scanning Calorimetry (DSC)

Thermal properties of the native starch and CLs (10% RL) samples were analyzed using a DSC (Norwalk, CT, USA) equipped with a refrigerated cooling system and Pyris^TM^ operation software (Perkin-Elmer), according to the former published method [15] with modifications. Starch samples (starch: water = 3:7, dry base) were weighed into a stainless-steel pan designed to withstand high pressure, an empty pan was used as the reference. Samples were equilibrated at 20 °C, and then heated to 100 °C at a scanning rate of 10 °C/min.

### 2.6. Pasting Properties

Pasting profiles of the native starch and CLs were determined using Micro Visco-Amylo-Graph (Brabender, Germany). The starch (6.00 g, dry base) was slurried in distilled water to reach a total weight of 100 g to prepare a starch suspension of 6% (*w*/*w*, dry base) [15]. Here in the pasting profiles, two pasting characteristic values (peak viscosity and final viscosity) and two operational values (breakdown and setback viscosities) are worth noting. Breakdown is the difference between peak viscosity and trough, and setback is the difference between final viscosity and trough [29].

### 2.7. Iodine-Binding Capacity (IBC)

The IBC of the native starch and CLs was measured according the previously reported method with modifications [21]. First, 100 ± 0.5 mg starch (dry base) was placed into a 15 mL centrifuge tube, then 1 mL ethanol added for sufficient dispersion of starch; a vortex mixer was applied while adding the NaOH (9 mL) to ensure a homogeneous solution. Then the above solution was equilibrated at room temperature for 10 min, then transferred into the 100 mL volumetric flask. After that, 5 mL of the above chemical gelatinized starch solution was transferred into another 100 mL volumetric flask (with 50 mL distilled water inside), 1 mL acetic acid solution (for neutralization, 1 M), and 2 mL I_2_-KI (configured according to the description of ISO approach) were added and made 100 mL for the color reaction. Thirty minutes later, the absorbance of the native starch (A) and the CLs (α) was measured by spectrophotometer. The blank solution was also configured according to ISO. The CL% was determined by the following Equation (1):CL% = (A − α)/A × 100%(1)
where CL% is the cross-linking degree calculated by the colorimetric method mentioned above, A is the absorbance of the native starch, and α is the absorbance of the CLs.

### 2.8. Statistical Analysis

Statistical differences among the data of native and cross-linked starches were analyzed using one-way ANOVA and Fisher LSD multiple comparisons. *p* values < 0.05 were deemed to be statistically significance. The statistical analyses were conducted on IBM SPSS Statistics Analysis (Version 26, IBM Corporation, Armonk, NY, USA). To control data quality, all tests were duplicated.

## 3. Results and Discussion

### 3.1. Cross-Linking Mechanisms of Normal and Waxy Starches

As shown in Figure 1, NM, WM, NP, and WP, differing in amylose content, were cross-linked to make comparisons and to determine their contributions to the gelatinization and pasting behavior of amylose on the one hand, and the crystalline pattern on the other. During cross-linking, amylose–amylopectin and amylopectin–amylopectin linkages were constructed in normal starches, but only the amylopectin–amylopectin linkage was constructed in waxy starches. After cross-linking, some of the amylopectin side chains were cross-linked together, and the uncross-linked side chains could provide space for iodine iodide, which stained brown, meaning the brown intensity decreased with the increase of the CL%.

### 3.2. XRD Pattern

The maize starches, waxy or normal, with or without cross-linking, displayed a typical A-type crystalline structure with the diffraction peaks at 15°, 17°, 18°, and 23°, while all the potato starches displayed a typical B-type crystalline structure at 5.6°, 17°, 22°, and 24° (Figure 2). After cross-linking, the effect of cross-linking on maize starch was little, similar to what was previously described by Liu et al. [30] and Park et al. [31], who have found that starch phosphorylation by STMP had no significant effect on the XRD pattern of NM and WM, respectively. For potato starches, Lemons et al. [32] and Wang et al. [33] have found NP and WP showed a decreased crystallinity after cross-linking. However, this study displayed remarkably different results after cross-linking, with a decrease in the relative crystallinity of WP and an increase in NP. As depicted in Figure 2, the intensity of the peaks decreased, especially at 2θ = 5.6° and 17°, and these peaks also gradually widened. The effect of the increase in peak width was higher than the decrease of peak height and, as a result, the peak area of 10 NP increased. Together with the DSC results (Table 1 and Figure 4), the onset gelatinization temperature of NP was the lowest of all samples, indicating that NP was more prone to partial gelatinization from the a structural rearrangement resulting from alkalinization from phosphorous cross-linking [32].

### 3.3. Morphology of Starch Granules

The images in Figure 3 showed that the maize starches (normal and waxy) of native and CLs were round and polygonal. A typical dark cross was shown in the polarized pictures, which agrees with earlier observations [21,34]. The potato starches (normal and waxy) of native and CLs showed a round and ellipsoidal granular shape with smooth surfaces. All the potato starch granules appear as round or distorted spherical crystals with a typical dark cross, with the distorted spherical large granules and small round granules. The native starch and its CLs did not have much variation in the morphology of granules, this was consistent with previous studies [35,36], where the cross-linked starches showed no significant change after modification by acetylated adipate, epichlorohydrin, phosphoryl chloride, and STMP/STPP. However, citric acid cross-linked starch exhibited an irregular and distorted shape with a groovy and rough surface [20]. This is mainly due to the different cross-linking mechanisms; the former is that the starch molecules are cross-linked when they are opened by NaOH, while citric acid could simultaneously destruct the crystalline structure and chemically cross-link the molecules. More interestingly, as all the pictures of the starch samples were taken under the same settings, the potato starch granules, which showed a B crystalline pattern, were much larger in size and brighter in their birefringence pattern than the maize starch granules. The possible reason for this is that the potato starch has a higher relative crystallinity by its longer side chains [26]. The waxy starches were found to have a slightly stronger birefringence intensity, which may be attributed to the disturbing effect of the amylose on normal starches [37].

### 3.4. Thermal Analysis

The gelatinization endotherm of the starch–water system of the native and 10% CLs of different starches at a temperature range between 30 and 100 °C was provided in Figure 4, and the gelatinization temperatures (*T_o_*, onset temperature; *T_p_*, peak temperature; *T_c_*, conclusion temperature) and enthalpy (△H) were shown in Table 1. The enthalpy values of the four starches followed the order: WP > WM > NM > NP. The potato starches displayed a lower *T_o_*, *T_p_*, and *T_c_* than maize starches, and waxy starches had higher △H than normal starches, which was consistent with the results of previous studies [38,39]. After cross-linking, the *T_o_*, *T_p_*, *T_c_*, and △H of NM increased significantly. The *T_o_* of NP also increased significantly, but the effect of cross-linking on WM and WP was not obvious. Generally, cross-linking reinforced the intramolecular hydrogen bonds [40]. In normal starches, amylose was cross-linked to amylopectin, and amylopectin was also cross-linked to amylopectin [15], but there was no linkage among the amylose molecules (cross-linking starch in granular form) [41]. However, amylose–amylopectin was to be reported thermally unstable [15]. In conclusion, the joint effect of reinforcing hydrogen bonds and thermal-unstable amylose–amylopectin increased the △H of normal starch.

### 3.5. Pasting Properties

WP showed a higher peak viscosity (384 BU) than NP (340 BU), which was different from a previous published paper where the peak viscosity of NP was higher than WP [42]. In that paper, viscosity analysis was performed using an 8% starch dry base, whereas here, we used 6%. In the concentrated regime, the viscosity was governed by particle rigidity [43], suggesting the higher rigidity of NP. Some CLs with a high RL were cross-linked too much to develop a detectable viscosity, NM stopped at 0.1%, NP at 0.5%, and WM at 1%. In addition, WP could develop viscosity even at 10%, albeit being less viscous than native starch (Table 2 and Figure 5). Increased intramolecular cross-linking due to a high cross-linking density might mean that self-aggregation may occur. Of all the starches, 1 WP had a significantly higher final viscosity (1270 BU), and its pasting curve increased from the beginning of gelatinization all through the heating and cooling process, and it finally gave a jagged profile. The jagged pasting profile was also observed in a previous study of WP and WM, and a possible cause could be the highly stringy paste of the starches [42]. Viscosity is also a method to characterize the CL%, but it cannot characterize the highly cross-linked starches as their viscosity is undetectable [21]. The peak viscosity increased from 96 BU (NM) to 108 BU (0.1 NM), an increase from 340 BU (NP) to 419 BU (0.1 NP) was observed for normal starches, meanwhile, it also increased from 237 BU (WM) to 551 BU (0.1 WM), and from 384 BU (WP) to 1163 BU (0.5 WP) for waxy starches. It can be concluded that the peak viscosity increment of waxy starch was more insignificant, and the effect of cross-linking was more significant on waxy starch and B-crystalline starches. It is worth noting that 0.01 NP showed a lower peak viscosity but a higher final viscosity than NP, and this phenomenon could also be observed in 0.01 NM.

Breakdown is caused by shear thinning, and the higher breakdown is attributed to the fact of a lower mechanical strength. After cross-linking, the breakdown values of NM decreased significantly, indicating the increase of mechanical properties [36,44]. Compared with normal starches, waxy starches showed a higher breakdown, and compared with A crystalline maize starches, B crystalline potato starches showed a higher breakdown. Setback and final viscosity correlated highly with the amylose content, it also indicated the starch retrogradation during the cooling period [10]. Setbacks positively correlated with amylose and longer branch chains [39,45]. Normal starches and B crystalline potato starch typically had greater setback viscosities, such as 68 BU of NM over 30 BU of WM, and 108 BU of NP over 32 BU of WP. The setback and final viscosity fluctuated with the peak viscosity, and there was no significant direct correlation to the RL of the cross-linker. This can also be certificated from another research on the CLs of wheat starch, where the setback and final viscosity also fluctuated with the peak viscosity [46].

### 3.6. Iodine-Binding Capacity

The normal starch–iodine complexes were blue in color and the waxy starch–iodine complexes were brown in color. After cross-linking, the color intensity faded with the increase of the cross-linker (Figure 6). This was also the theoretical basis for the new method of CL% characterization. Figure 6 also presented a linear-log plot of the CL% vs. the RL of the cross-linker, the linear regression to the semi-log plot were also shown (dotted lines), and the R-squared ranged approximately from 0.79 (WP) to 0.98 (NP and WM). The line fit through the data was better, but the R-squared of the WP line was low (0.79). When the starch was cross-linked, amylopectin was more inclined to be cross-linked than amylose, and no cross-link was found among the amylose molecules [41,47]. As a result, the full amylopectin WM cross-linked faster, and meanwhile, its CL% was higher than that of NM. In addition, the WM, NP, and WP lines showed a bigger slope, suggesting the higher efficiency of the cross-linking reaction.

## 4. Conclusions

The STMP/STPP cross-linking reaction was a modification at the molecular level, and there was little difference in the morphology between the native and the CLs of the four starches. Amylopectin was more prone to cross-linking and the cross-linked waxy starches displayed significantly higher viscosities than normal starches. B-crystalline potato starch was more susceptible to cross-linkers, which showed higher viscosities than A-crystalline maize starch: 580 BU of 0.1 WM over 181 BU of 0.1 NM, and 929 BU of 0.1 WP over 534 BU of 0.1 NP. The final viscosity of 1 WP was the highest of all samples at 1270 BU. Waxy starches had higher gelatinization enthalpy due to their higher cross-linking degree, the difference between WP and NP was higher than that of WM and NM. Most importantly, the simple and high-performance St-I method can be applied to characterize the cross-linking degree of the waxy starches of different RLs. In particular, the CL% and RL showed width distributions following linear-log functions. Combined with the pasting data St-I method, it had a high sensitivity and a wide range of responses.

## Figures and Tables

**Figure 1 polymers-14-02870-f001:**
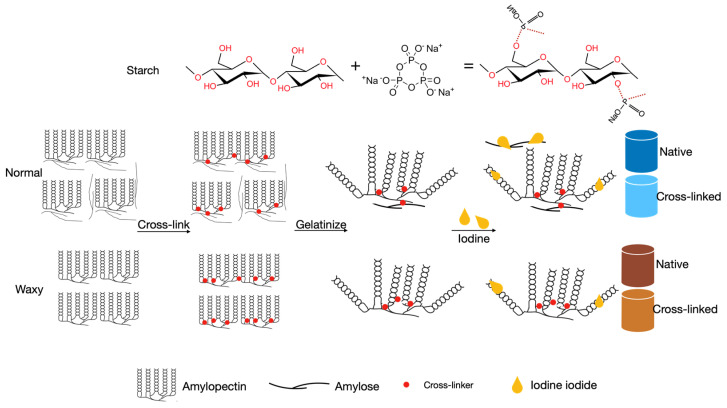
Different mechanisms of cross-linking reaction between normal and waxy starches.

**Figure 2 polymers-14-02870-f002:**
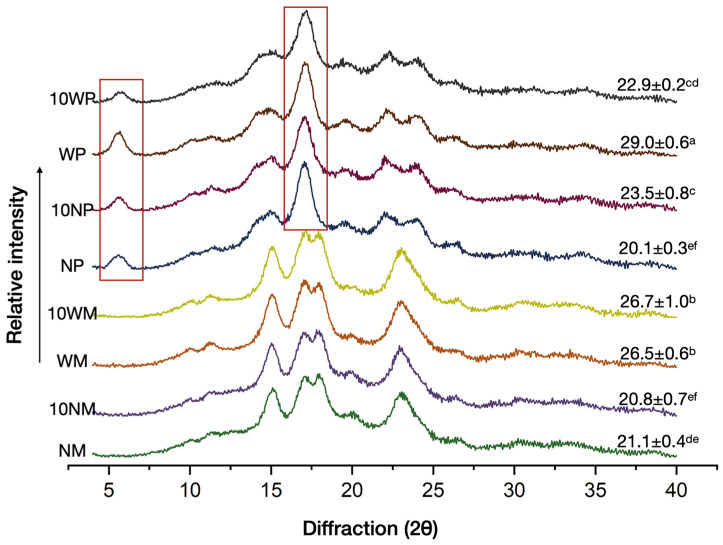
XRD patterns and relative crystallinity (%) of native and CLs prepared with a 10% reagent level. NM, WM, NP, and WP are normal maize, waxy maize, normal potato, waxy potato; 10 NM, 10 WM, 10 NP, and 10 WP are the cross-linked starches.

**Figure 3 polymers-14-02870-f003:**
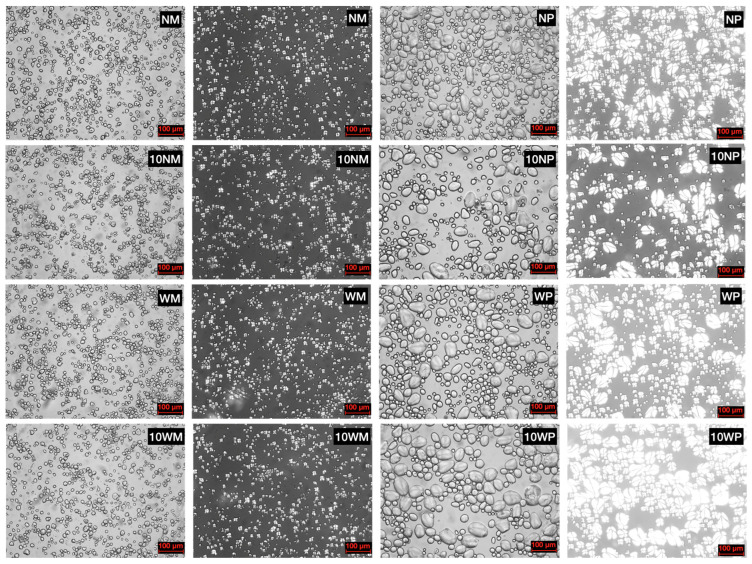
Light microscope of native and CLs prepared with a 10% reagent level (scale bar = 100 μm). Black background pictures are the same field as the grey background taken under polarized light.

**Figure 4 polymers-14-02870-f004:**
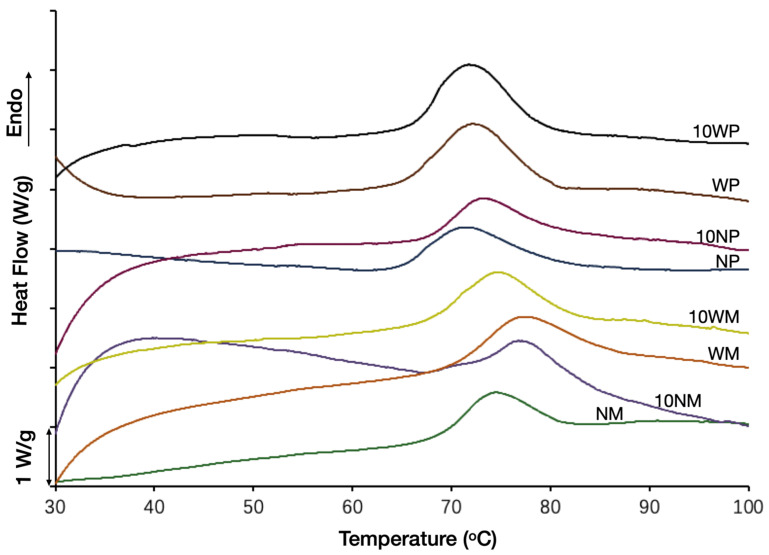
Gelatinization endotherms of native and CLs prepared with a 10% reagent level.

**Figure 5 polymers-14-02870-f005:**
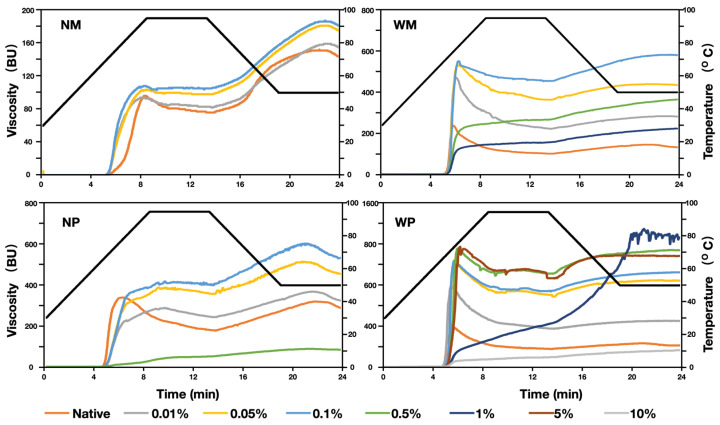
Pasting properties of native starch and cross-linked starches analyzed using a Micro Visco-Amylo-Graph. NM, WM, NP, and WP are normal maize, waxy maize, normal potato, and waxy potato.

**Figure 6 polymers-14-02870-f006:**
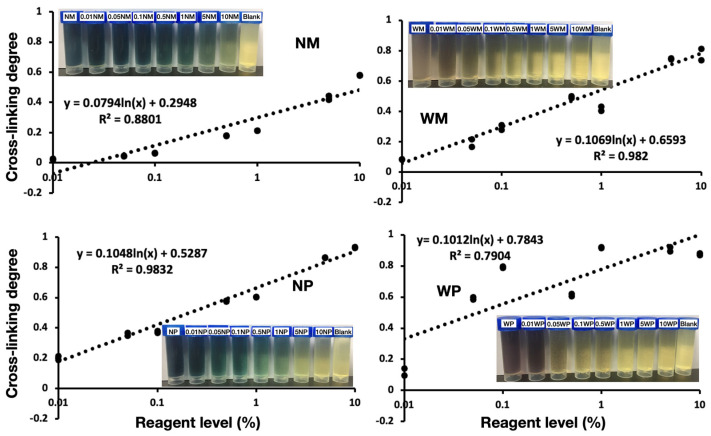
Cross-linking degree of cross-linked starches through iodine-binding capacity. NM, WM, NP, and WP are normal maize, waxy maize, normal potato, and waxy potato.

**Table 1 polymers-14-02870-t001:** Thermal properties of native and CLs of 10% reagent level.

Samples	*T_o_* (°C)	*T_p_* (°C)	*T_c_* (°C)	△H (J/g)
NM	68.01 ± 0.93 ^bc^	72.64 ± 0.89 ^cde^	78.52 ± 0.16 ^d^	14.53 ± 0.18 ^d^
10 NM	71.64 ± 0.71 ^a^	77.30 ± 0.32 ^a^	83.45 ± 0.06 ^b^	16.23 ± 0.55 ^c^
WM	69.13 ± 0.83 ^b^	75.18 ± 0.08 ^ab^	86.09 ± 0.85 ^a^	18.51 ± 0.59 ^b^
10 WM	67.93 ± 0.39 ^bc^	74.56 ± 0.98 ^bc^	81.85 ± 0.71 ^c^	17.67 ± 0.22 ^bc^
NP	65.69 ± 0.26 ^d^	71.33 ± 0.86 ^e^	78.33 ± 0.33 ^d^	12.47 ± 0.53 ^e^
10 NP	67.88 ± 0.08 ^bc^	73.20 ± 0.67 ^bcde^	79.73 ± 0.62 ^d^	13.4 ± 0.35 ^de^
WP	65.41 ± 0.03 ^d^	72.18 ± 0.85 ^e^	78.72 ± 0.19 ^d^	21.93 ± 0.60 ^a^
10 WP	66.55 ± 0.73 ^cd^	71.76 ± 0.10 ^e^	78.72 ± 0.04 ^d^	21.53 ± 0.50 ^a^

**Table 2 polymers-14-02870-t002:** Pasting parameters for native and CLs of different reagent levels (6% starch, db).

Samples	Peak η ^1^/BU	Final η/BU	Break-Down/BU	Set-Back/BU	Samples	Peak η/BU	Final η/BU	Break-Down/BU	Set-Back/BU
NM	96 ± 2.0 ^bc^	144 ± 4.5 ^c^	20 ± 3.0 ^a^	68 ± 5.5 ^a^	WM	237 ± 4.0 ^d^	133 ± 4.5 ^f^	134 ± 3.0 ^c^	30 ± 3.5 ^e^
0.01 NM	94 ± 2.5 ^c^	155 ± 1.5 ^b^	11 ± 0.5 ^b^	72 ± 4.0 ^a^	0.01 WM	472 ± 5.5 ^b^	282 ± 4.5 ^d^	247 ± 2.0 ^a^	57 ± 3.0 ^d^
0.05 NM	104 ± 3.0 ^ab^	175 ± 4.5 ^a^	6 ± 2.0 ^bc^	77 ± 2.0 ^a^	0.05 WM	531 ± 7.0 ^a^	434 ± 7.0 ^b^	167 ± 3.5 ^b^	70 ± 2.0 ^c^
0.1 NM	108 ± 1.0 ^a^	181 ± 1.5 ^a^	3 ± 1.0 ^c^	76 ± 1.0 ^a^	0.1 WM	551 ± 13 ^a^	580 ± 11 ^a^	95 ± 2.0 ^d^	124 ± 4.0 ^a^
					0.5 WM	267 ± 8.0 ^c^	365 ± 2.5 ^c^	0	98 ± 1.5 ^b^
					1 WM	156 ± 2.5 ^e^	224 ± 3.0 ^e^	0	68 ± 1.0 ^c^
NP	340 ± 7.5 ^c^	289 ± 3.0 ^d^	159 ± 1.5 ^a^	108 ± 4.5 ^b^	WP	384 ± 8.0 ^d^	212 ± 5.5 ^g^	204 ± 4.5 ^e^	32 ± 1.5 ^e^
0.01 NP	288 ± 3.0 ^d^	326 ± 2.5 ^c^	42 ± 4.5 ^b^	80 ± 1.5 ^d^	0.01 WP	795 ± 12 ^c^	453 ± 7.0 ^f^	415 ± 5.0 ^a^	73 ± 3.5 ^d^
0.05 NP	388 ± 2.0 ^b^	455 ± 3.0 ^b^	29 ± 3.0 ^bc^	96 ± 2.5 ^c^	0.05 WP	1037 ± 9.5 ^b^	848 ± 5.0 ^e^	327 ± 8.5 ^b^	138 ± 2.5 ^c^
0.1 NP	419 ± 9.0 ^a^	534 ± 5.0 ^a^	17 ± 3.5 ^c^	132 ± 1.5 ^a^	0.1 WP	1042 ± 14 ^b^	929 ± 8.0 ^d^	296 ± 5.0 ^c^	183 ± 3.5 ^b^
0.5 NP	51 ± 1.5 ^e^	83 ± 2.5 ^e^	0 ^d^	32 ± 1.5 ^e^	0.5 WP	1163 ± 17 ^a^	1146 ± 12 ^b^	246 ± 1.0 ^d^	229 ± 5.0 ^a^
					1 WP	421 ± 8.0 ^d^	1270 ± 13 ^a^	nd ^2^	nd
					5 WP	1180 ± 18 ^a^	1090 ± 10 ^c^	308 ± 7.0 ^bc^	218 ± 7.5 ^a^
					10 WP	97 ± 4.0 ^e^	166 ± 3.5 ^h^	0	69 ± 1.0 ^d^

^1^ η = viscosity. ^2^ nd = not detected. 0.01 NM, 0.05 NM, and 0.1 NM: normal maize was cross-linked with 0.01%, 0.05%, and 0.1% reagent level. WM, NP, and WP are waxy maize, normal potato, and waxy potato.

## Data Availability

The data that support the fundings of this study are available from the corresponding author upon reasonable request.

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
