# Peer review of "Effect of Amylose and Crystallinity Pattern on the Gelatinization Behavior of Cross-Linked Starches"

_polymers, 2022, doi:10.3390/polym14142870_

Round 1

Reviewer 1 Report

The authors have done a good job in revisiting the manuscript and making it better by answering most of the queries raised. However, they failed to address some of the below-given comments. I recommend its publication after addressing the below-mentioned comments:

1. The DSC curves should be presented in the manuscript instead of supplementary information. The quality of the figure needs to be improved, e.g. the figure must contain a y-axis and a correct temperature range. Moreover, the figure caption should describe the exact analysis namely DSC thermograms.      

2. The authors have presented the X-ray diffraction results in the supplementary information with no discussion and no explanation in the experimental section. They just repeated the discussion of 3.4. section (Pasting properties). Therefore, I highly recommend adding the XRD results altogether with its related discussion and experimental part to the manuscript since it is important to discuss the crystallinity of the starch samples. Please, improve the quality of the figure as well (adding a y-axis).

3. Still, there are many grammatical errors in the manuscript e.g. Line 67, 92, 198, 212-214, 253, and many other places. The manuscript requires extensive editing of the English language and style.

Author Response

  1. The DSC curves should be presented in the manuscript instead of supplementary information. The quality of the figure needs to be improved, e.g. the figure must contain a y-axis and a correct temperature range. Moreover, the figure caption should describe the exact analysis namely DSC thermograms.      

We have put the DSC curves in the manuscript, and Y-axis was included, figure caption was edited.

  1. The authors have presented the X-ray diffraction results in the supplementary information with no discussion and no explanation in the experimental section. They just repeated the discussion of 3.4. section (Pasting properties). Therefore, I highly recommend adding the XRD results altogether with its related discussion and experimental part to the manuscript since it is important to discuss the crystallinity of the starch samples. Please, improve the quality of the figure as well (adding a y-axis). 

XRD results was put in the result and discussion part.

  1. Still, there are many grammatical errors in the manuscript e.g. Line 67, 92, 198, 212-214, 253, and many other places. The manuscript requires extensive editing of the English language and style.

We have re-write all the mentioned part carefully, and the language of the manuscript was also edited.

Reviewer 2 Report

The manuscript has investigated the influence of amylose and crystallinity pattern on the gelatinization behavior of cross-linked starches. Several papers have investigated this topic and the study does not really contribute new additional knowledge.

The manuscript is poorly written and is not deeply discussed. Additionally, statistical analysis is not performed. The manuscript is not acceptable in its present form

Other comments:

 Crystallinity is written in the title, but the XRD data are presented in the supplementary file!

Lines 17-18: What do you mean by disappear?

Lines 66-82: many sentences are repeated several times. Please re-write this part.

Please add the allowed degree of substitution of cross-linked starches to the abstract.

Materials and methods: why you have not performed the tests for all of the treatments?

Line 122: “100 ± 0.5 mg (dry base)” is it the weight of starch? If so, please write “100 ± 0.5 mg starch (dry base)”

Lines 130-131: was measured by what? A spectrophotometer? If so, please add the name and model of the instrument.

Why you have not performed a statistical analysis?

Please write how many times each test was replicated.

Line 146: The red …

Figure 2: the resolution of image 2 is not high. The scale bar is not clear

Line 156: images in Fig

Lines 181-183: you cannot make a conclusion without statistical analysis.

Data presented in the tables are not analyzed

Lines 206 and line 224: how did you understand they have significant differences without statistical analysis?!

Table 2: please add the standard deviations

Lines 224-226: the statements do not match the presented data

Author Response

The manuscript has investigated the influence of amylose and crystallinity pattern on the gelatinization behavior of cross-linked starches. Several papers have investigated this topic and the study does not really contribute new additional knowledge.

We systematically studied four starches cross-linked with STMP/STPP by 7 different concentrations (0.01, 0.05, 0.1, 0.5, 1, 5, 10%), which has not been reported before.

The manuscript is poorly written and is not deeply discussed. Additionally, statistical analysis is not performed. The manuscript is not acceptable in its present form

Thanks for the remarks, and we have added statistical analysis.

Other comments:

  1. Crystallinity is written in the title, but the XRD data are presented in the supplementary file!

XRD results was put in the result and discussion part.

  1. Lines 17-18: What do you mean by disappear?

No viscosity can be detected, and we have changed the statement.

  1. Lines 66-82: many sentences are repeated several times. Please re-write this part.

We have re-write this part, many thanks to the suggestions.

  1. Please add the allowed degree of substitution of cross-linked starches to the abstract.

For food and drug industries, there are limitations on the starches cross-linked by STMP/STPP, and we have measured the phosphate content in a previous paper (DOI: 10.1016/j.carres.2018.01.009), the phosphate content of 10% cross-linked NM and NP was significantly lower than 0.4%.

According to FDA, when used STMP in starch modification, residual phosphate in food starch-modified not to exceed 0.04 percent, calculated as phosphorus. When used STMP/STPP, residual phosphate in food starch-modified not to exceed 0.4 percent calculated as phosphorus.

FDA. U.S. Food and Drug Administration, CFR - Code of Federal Regulations Title 21, Avaliable at: https://www.accessdata.fda.gov/scripts/cdrh/cfdocs/cfCFR/CFRSearch. cfm?fr=172.892 2016 (Access: June, 2022)

  1. Materials and methods: why you have not performed the tests for all of the treatments?

In this paper, our purpose is to characterize the cross-linking degree of waxy starches by a simple method, and compare the differences between waxy and normal starches, A- and B-crystalline starches. DSC, XRD and light microscopy were used to study the effect of crosslinking to the starches, as your remarks above “Several papers have investigated this topic and the study does not really contribute new additional knowledge”. In addition, if we put all the images of the samples, there would be 64 pictures, 32 DSC and XRD lines, which will take too much spaces of the manuscript.

  1. Line 122: “100 ± 0.5 mg (dry base)” is it the weight of starch? If so, please write “100 ± 0.5 mg starch (dry base)”

Yes, thanks.

  1. Lines 130-131: was measured by what? A spectrophotometer? If so, please add the name and model of the instrument.

Yes, thanks.

  1. Why you have not performed a statistical analysis?

We have added the statistical analysis.

  1. Please write how many times each test was replicated.

All tests were duplicated.

  1. Line 146: The red …

The brown…

  1. Figure 2: the resolution of image 2 is not high. The scale bar is not clear

We have doubled the scale bar.

  1. Line 156: images in Fig

Yes, thanks.

  1. Lines 181-183: you cannot make a conclusion without statistical analysis.

After cross-link, the To, Tp, Tc and  of NM increased significantly, the To of NP also increased significantly. However, the effect of cross-linking on WM and WP was not obvious.

  1. Lines 206 and line 224: how did you understand they have significant differences without statistical analysis?!

We have added the statistical analysis to table 2.

  1. Table 2: please add the standard deviations

The standard deviations were added.

  1. Lines 224-226: the statements do not match the presented data

Compared with normal starches, waxy starches showed higher breakdown, and compared with A crystalline maize starches, B crystalline potato starches showed higher breakdown.

Breakdown of NM and NP are 20 BU and 159 BU, breakdown of WM and WP are 134 BU and 204 BU, respectively.

We think the statement match the presented data.

Round 2

Reviewer 1 Report

The authors have done a great job in revising the manuscript and improving it by answering the queries raised. Thus, I recommend the acceptance of the manuscript for publication in Polymers.

Author Response

Thank you!

Reviewer 2 Report

Line 256: Please correct the sub-heading "3.6. Iodine binding capacity"

Author Response

Line 256: Please correct the sub-heading "3.6. Iodine binding capacity"

Yes and thanks.

This manuscript is a resubmission of an earlier submission. The following is a list of the peer review reports and author responses from that submission.

Round 1

Reviewer 1 Report

In this manuscript, the authors investigated the effect of amylose and crystallinity patterns on the gelatinization behavior and the crosslinking reaction. The manuscript presents interesting results but needs substantial improvement before it can be considered for publication in Polymers, as I can list some of them below:

1. The abstract should show the importance of the determination of crosslinking degree by iodine binding capacity measurement. Moreover, the authors must add more important results in this section as well.  

2. Line 185, "The native starch and its CLs did not have much variation in the morphology of granules, because the cross-linking is a reaction of molecular level". According to the literature, the morphology of native starch has changed after crosslinking with citric acid due to the destructive effect of the crosslinking agent on the crystalline structure part of the starch (ref.: Polymers for Advanced Technologies 31.6 (2020): 1256-1269). Please, consider this phenomenon, mention it, and discuss your results more.     

3. Introduction needs to be improved. Maybe, you can restructure this section. The references must be updated by at least 50% between 2017-2022 to show the work's originality and its importance nowadays. Moreover, new methods and strategies for crosslinking of starch as well as introducing various crosslinking agents namely epichlorohydrin, citric acid, etc. with proper references should be provided by the authors.

4. What is the amylose/amylopectin ratio of the starches used in this study? Because the authors have stated that they investigated the effects of amylose content and crystallinity as the important parameters on crosslinking degree.

5. Line 103, "In the past, CLs has been prepared by researchers for various applications. These include thickeners, drug delivery, and composite materials". Please, cite some new references for this statement.

6. Figure 4 just shows various mechanisms of crosslinking reaction. It should be presented in the first part of the results and discussion to explain clearly the expected reactions between starch and the crosslinking agent.

7. The work presents potential in the field of crosslinked starch. Maybe, for further works, chemical analysis such as FTIR, solid-state NMR, etc., might provide interesting results.

8. Let the table and figure legends be more informative (especially abbreviations and sample codes) to avoid the readers getting back to the methodology section to understand what we can see in the results.

9. There are many grammatical errors. Please have a thorough proofread of the entire manuscript for proper English usage. 

Reviewer 2 Report

In my opinion, the work does not contain significant scientific novelty

The introduction, although quite long, does not lead to the formulation of a scientific problem.

The paper does not contain experiments indicating the type and degree of crystallinity of the tested starches, which are necessary for results discussion.

No DSC curves are given that support the data presented in Table 1.

The authors confuse the term "colorimetric" and "calorimetric".

It is difficult to talk about the larger or smaller sizes of the starch granules shown in Figure 1 because there is no size statistic.

Ad. Point 2.7, p.4 -  How can you do a statistical analysis of the results of the measurements that were only duplicate?

Changes in the chemical structure caused by chemical or heat cross-linking have not been confirmed by spectroscopic methods.

Due to the reported comparative viscosity properties of the starches tested, they could be presented in a journal with an application or technological profile.